## Research Article

diversification; fossils; phylogenies; Cenozoic

**Corresponding author:**
Bethany J. Allen;
Email: bethany.allen@bsse.ethz.ch

Tanja Stadler, Timothy G. Vaughan and Rachel C. M. Warnock contributed equally to this work.

# Mechanistic phylodynamic models do not provide conclusive evidence that non-avian dinosaurs were in decline before their final extinction

Bethany J. Allen[1,2] (iD), Maria V. Volkova Oliveira[3], Tanja Stadler[1,2], Timothy G. Vaughan[1,2] and Rachel C. M. Warnock[4]

[1]Department of Biosystems Science and Engineering, ETH Zurich, Basel, Switzerland; [2]Computational Evolution Group, Swiss Institute of Bioinformatics, Lausanne, Switzerland; [3]Independent and [4]Geozentrum Nordbayern, Friedrich-Alexander-Universität, Erlangen, Germany

## Abstract

Phylodynamic models can be used to estimate diversification trajectories from time-calibrated phylogenies. Here we apply two such models to phylogenies of non-avian dinosaurs, a clade whose evolutionary history has been widely debated. Although some authors have suggested that the clade experienced a decline in diversity, potentially starting millions of years before the end-Cretaceous mass extinction, others have suggested that the group remained highly diverse right up until the Cretaceous-Paleogene (K-Pg) boundary. Our results show that model assumptions, likely with respect to incomplete sampling, have a large impact on whether dinosaurs appear to have experienced a long-term decline or not. The results are also highly sensitive to the topology and branch lengths of the phylogeny used. Developing comprehensive models of sampling bias, and building larger and more accurate phylogenies, are likely to be necessary steps for us to determine whether dinosaur diversity was or was not in decline before the end-Cretaceous mass extinction.

## Impact statement

Dinosaurs are well known for their abrupt demise at the end of the Cretaceous period, coincident with the Chicxulub asteroid impact at 66 Ma. However, their diversity dynamics over the course of their preceding 180-million-year history are less well understood. It is not known, for instance, whether dinosaurs were thriving or already in decline just before the impact event. This is in large part due to their highly fragmentary fossil record. Phylogenetic trees depicting evolutionary relationships provide additional information, including capturing a portion of lineage history that is otherwise not observable from fossil occurrence data. Previous analyses based on dinosaur phylogenies have reached conflicting conclusions about the evolutionary trajectory of dinosaurs before their final extinction. Here, we revisit this conflict using a phylodynamic modelling approach, which is more explicit and transparent than previous approaches, especially with respect to the assumptions made about how the dinosaur fossil record has been sampled. Using two alternative models, which differ in how they use information about the sampling process and how they model changes in the number of species through time, we show that based on available phylogenies we cannot currently reach a definitive conclusion about dinosaur diversification during the Cretaceous. More densely-sampled and accurate fossil timetrees, as well as models that capture more information about the quality of the dinosaur fossil record, may help to solve this debate.

## Introduction

Dinosaurs were the dominant land animals of the Mesozoic, renowned for their diversity, disparity, and ecological novelty, but they are now represented by a single surviving subclade, birds (Brusatte et al., 2015; Benson, 2018). The extinction of non-avian dinosaurs at the end of the Cretaceous period (approximately 66 Ma) is widely accepted to be coincident with, and likely caused by, an asteroid impact (Alvarez et al., 1980; Schulte et al., 2010; Chiarenza et al., 2020; Hull et al., 2020). However, the trajectory of dinosaur diversity throughout the Mesozoic, especially towards the end of the Cretaceous, remains controversial. A wide variety of methods have previously been used to estimate either the number of dinosaur species or their diversification rates, including interpolation or extrapolation (Fastovsky et al., 2004; Wang and Dodson, 2006; Lloyd et al., 2008; Brusatte et al., 2015; Close et al., 2018) and modelling (via regression (Barrett

et al., 2009; Lloyd, 2011; Sakamoto et al., 2016; Bonsor et al., 2020), species-area relationships (Russell, 1995; Le Loeuff, 2012), or inferring evolutionary and/or sampling rates (Starrfelt and Liow, 2016; Condamine et al., 2021; Černý et al., 2022; Langer and Godoy, 2022)). Several papers have suggested that the group was already in decline before the asteroid impact (Han et al., 2022), and had been for the preceding 10 million years (Sloan et al., 1986; Archibald, 2014; Condamine et al., 2021), 24 million years (Sakamoto et al., 2016), or even the whole Cretaceous (Lloyd, 2011). However, others have argued that this was not the case and that dinosaurs remained highly diverse right up until the latest Cretaceous (Fastovsky et al., 2004; Wang and Dodson, 2006; Brusatte et al., 2015; Starrfelt and Liow, 2016; Bonsor et al., 2020). This debate sits within the context of approximately constant terrestrial tetrapod species richness throughout the Mesozoic (e.g. Benson et al., 2016; Close et al., 2017). Several possible drivers of a Cretaceous dinosaur decline have been put forward, such as environmental change resulting from Deccan Traps volcanism or sea level fluctuations, however, the poor temporal resolution of geological records at this time has hindered efforts to correlate potential causes and effects (Brusatte et al., 2015; Benson, 2018; Chiarenza et al., 2019, 2020). There is evidence of trophic restructuring in the latest Cretaceous, which may have left Maastrichtian food webs more vulnerable to perturbations (Mitchell et al., 2012; García-Girón et al., 2022).

Many previous studies have commented on variability in the sampling of the dinosaur fossil record, across space, time, and clades (Wang and Dodson, 2006; Barrett et al., 2009; Benton et al., 2011; Upchurch et al., 2011; Benson, 2018; Chiarenza et al., 2019; Cashmore et al., 2020; Dean et al., 2020). This hinders efforts to accurately estimate species richness or diversification over geological time, as any true changes in diversity are likely to be obscured by sampling bias (e.g. Starrfelt and Liow, 2016; Benson et al., 2021). The various methods that have been applied to estimating dinosaur diversity handle this information differently: some simply take the fossil record at face value and assume that any potential sampling biases are negligible, whereas others explicitly model the incompleteness of the fossil record and thereby infer what we do not know from the fossils we have. An example of this latter viewpoint is Chiarenza et al. (2019), who used ecological niche modelling to infer where dinosaurs could have lived during the Late Cretaceous based on their environmental preferences, extrapolating beyond the area represented by known fossil occurrences.

The diversification history of a clade can be quantified using raw fossil occurrences (via approaches such as PyRate (Silvestro et al., 2014; Condamine et al., 2021; Černý et al., 2022; Langer and Godoy, 2022)) and phylogenetic trees (e.g. Černý et al., 2022; Langer and Godoy, 2022; Truman et al., 2024). Although the fossil record contains key information about the presence of taxa at a specific place and time in the geological past, this information is highly patchy, whereas phylogenies have the advantage of capturing a portion of evolutionary history that is not directly observable (e.g. Lloyd et al., 2008; Starrfelt and Liow, 2016; Benson, 2018). Tree shape, in particular the temporal distribution of node ages and branch lengths, is informative about patterns of diversification, and provides insight into parts of the tree of life which are not currently represented within the known fossil record (Lloyd et al., 2008).

A handful of studies have previously used phylogenies to test whether dinosaur diversity was in decline before the Cretaceous-Paleogene (K-Pg) mass extinction. One approach is using phylogenetic generalised linear mixed models (GLMMs), which assess the line of best fit between the time elapsed from the root to the tips (the predictor variable) and net speciation (the response variable).

The shape and slope of this line can be used to infer whether diversity has remained constant, increased, or decreased over time. Sakamoto et al. (2016) applied phylogenetic GLMMs to three dinosaur supertrees and found evidence of a diversity decline, starting at least 24 million years before the end-Cretaceous mass extinction. However, subsequent discourse (Bonsor et al., 2020; Sakamoto et al., 2021) has raised questions about the "correct" way to apply this method, including how best to interpret mixed results and how sensitive the method is to the shape of the phylogeny. The adequacy of model fit can also be problematic (Hadfield, 2010); in part, this may occur because the method fits a single smoothed curve to the entirety of the clade's evolutionary trajectory, which does not allow for short-term fluctuations in rates to be recovered. Sakamoto et al. (2016) also attempted to account for sampling bias by including geological and sampling proxy data as covariates in their phylogenetic GLMMs, and found that this did not change their overall results. However, this approach does not incorporate the sampling process explicitly or formalise the relationship between diversification and sampling (Bonsor et al., 2020; Warnock et al., 2020). As a result, the effect of incomplete fossil sampling on phylogenetic GLMMs is difficult to assess.

To examine the potential impact of modelling assumptions on estimates of diversification from the non-avian dinosaur fossil record, we apply Bayesian phylodynamic models. Although phylogenetics describes the process of inferring evolutionary relationships, phylodynamics seeks to infer characteristics of the history of the clade, such as diversification rates or diversity through time (Grenfell et al., 2004). Here, we use two different phylodynamic models, which make different assumptions about sampling and changes in the number of species through time, to infer dinosaur diversification over the Mesozoic. Both models generate piecewise-constant trajectories, allowing parameters to be estimated within a series of predefined time intervals. The first, a coalescent model, conditions the diversification process on the observed fossil ages, treating each sample as an independent event, whereas the second, a birth-death-sampling model, instead models sampling as an explicit process that generates the observed fossil record. The number of species through time changes deterministically under the coalescent model, but under the birth-death model, this change is stochastic. In the manner by which both sampling and species numbers are treated, the coalescent model is more similar to phylogenetic GLMMs. Our results show that phylodynamic models do not conclusively support a decline in dinosaur diversity towards the end of the Cretaceous, and indicate that accurately modelling sampling bias is likely to be key to understanding diversification dynamics in deep time.

## Methods

### *Phylogenies*

We used four dinosaur supertrees, the same three as Sakamoto et al. (2016) in addition to a more recently constructed "metatree" (Lloyd et al., 2017). To create a fully bifurcating topology for the metatree, we sampled 1,000 phylogenies from the set of most parsimonious trees and generated a maximum clade credibility tree using TreeAnnotator (Rambaut and Drummond, 2021).

To infer the branch lengths of the phylogenies, age range data for all non-avian dinosaur species were downloaded from the Paleobiology Database (Uhen et al., 2023) in December 2022, with species names then matched to the tip names in the phylogenies (modifications are described in the electronic supplement). Any informal species, birds (*Archaeopteryx* and more bird-like species),

and species without age information were removed from the phylogenies, using the ape (Paradis and Schliep, 2019) and palaeoverse (Jones et al., 2023) packages in R (R Core Team, 2022). Following this cleaning, the smallest phylogeny contained 391 dinosaur species (Lloyd et al., 2008) (hereafter 'Lloyd1'), the two medium-sized phylogenies comprised the same 542 species but differed in their topologies (Benson et al., 2014) (hereafter 'Benson1' and 'Benson2'), and the largest phylogeny contained 750 species (Lloyd et al., 2017) (hereafter 'Lloyd2'). As well as analysing the supertrees in full, we also divided each into its three major subclades (Ornithischia, Theropoda, and Sauropodomorpha). We therefore conducted our analyses on a total of 16 phylogenies.

## Phylodynamic models

We used two distinct Bayesian phylodynamic models to infer diversification dynamics from these species trees: the birth-death skyline (BDSKY) model (Stadler, 2011; Stadler et al., 2013; Gavryushkina et al., 2014; Heath et al., 2014) and a piecewise-exponential population size model based on Kingman's $n$-coalescent process (Kingman, 1982; Griffiths and Tavaré, 1994). Although both models are often used in the analysis of epidemiological phylogenies, they are yet to be widely applied in macroevolution.

The BDSKY model assumes that all of the observed species are the result of a birth-death process that began with a single species at some unknown time in the past. It also assumes that time is divided into one or more intervals; here, we defined eight time bins based on different geological intervals (see below). Within a single interval $i$, species give rise to new species at the constant rate $\lambda_i$ (per co-existing species per Myr), and go extinct at a constant rate $\mu_i$ (per species per Myr). Additionally, fossils are produced at rate $\psi_i$ (per co-existing species per Myr). Species are not removed after sampling, allowing (in principle) sampled species to be direct ancestors of one another (Gavryushkina et al., 2014). As our phylogenies only include non-avian (extinct) dinosaurs, we assume no extant sampling ($\rho = 0$), and condition the model on producing at least one fossil. A diversification ($\lambda_i - \mu_i$) rate was calculated post-hoc for each interval in each iteration.

The piecewise-exponential coalescent model we use assumes that the observed tree is the result of a coalescent process parameterised by a time-dependent effective population size function, $N(t)$. At any given time, the value of this function can be interpreted as proportional to a number of extant species, and thus we also refer to it as the effective species richness. We assume that this function has a continuous, piecewise-exponential form, with growth rates in each interval given by the diversification rate parameter, $r_i$, together with the effective species richness at the end of the most recent interval (here, the Coniacian–Maastrichtian), $N_f$. A key difference between this model and the BDSKY model is that the coalescent does not explicitly model the sampling process; it simply assumes that the sample dates (fossil ages) are independent of the number of species over time, and that the species sampled are drawn randomly from all co-existing species.

We used the boundaries of eight geological intervals of approximately equal length (Early–Mid Triassic, prior to 237 Ma; Late Triassic, 201.4–237.0 Ma; Early Jurassic, 174.7–201.4 Ma; Middle Jurassic, 161.5–174.7 Ma; Late Jurassic, 145.0–161.5 Ma; Berriasian–Barremian, 121.4–145.0 Ma; Aptian–Turonian, 89.8–121.4 Ma; Coniacian–Maastrichtian, 66.0–89.8 Ma (Cohen et al., 2013)) as the change times for the piecewise rates in all models.

## Bayesian inference of model parameters

Our models constitute a specific hypothesis for how the empirical phylogenetic tree $T$ was produced, and are evaluated using the probability of observing this tree given the model-specific parameters. We used Bayesian inference to infer these model parameters, as well as the branch lengths, from the predetermined phylogenetic relationships in the supertrees and the imposed tip constraints.

Specifically, conditional on a phylogenetic tree $T$ and a particular phylodynamic model $M$, we seek to infer the model parameters $\Theta_M$. In the case that $M$ is the BDSKY model, $\Theta_{BDSKY} = \left\{ \vec{\lambda}, \vec{\mu}, \vec{\psi}, t_{or} \right\}$, whereas when $M$ is the coalescent model $\Theta_C = \left\{ \vec{N}, \vec{r} \right\}$. In the Bayesian context, inference amounts to characterisation of the posterior distribution

$$P(\Theta_M | T, M) = \frac{P(T|M, \Theta_M) P(\Theta_M|M)}{P(T|M)},$$

where $P(T|M, \Theta_M)$ is the likelihood of the model parameters given the tree under the particular model $M$, $P(\Theta_M|M)$ is the prior probability distribution for the model parameters, and $P(T|M)$ is the marginal likelihood of the model (which is constant with respect to the model parameters). In both the BDSKY and coalescent models, we express $P(\Theta_M|M)$ as a product of priors for each of the individual model parameters, meaning that we assume no correlation between these individual parameters.

The prior probability distributions used for the individual parameters are listed in Table 1. The scale of the birth, death, and diversification rate priors was based on estimates from a study calculating diversification rates in a large number of extant and extinct phylogenies (0.02 to 1.54 speciation/extinction events per lineage per million years) (Henao Diaz et al., 2019). In the absence of robust methods for estimating sampling completeness from the fossil record, our prior on the sampling rate favours smaller values (the mean represents one sample per lineage per 5 million years; Table 1) but does not explicitly exclude larger values.

The branch lengths of the phylogenies were also inferred within the BDSKY and coalescent analyses. Tip constraints were placed on each species, in the form of a uniform probability distribution ranging from the oldest possible age of the oldest fossil to the youngest possible age of the youngest fossil. In each MCMC iteration, the age of each fossil, together with internal node ages and phylodynamic parameters, was sampled. Through this joint inference, we take into account uncertainty in the branch lengths, with the origination and extinction times of each lineage occurring before and after the sampled fossil age, respectively. In this analysis, the K-Pg boundary is treated as analogous to the "present day"

**Table 1.** Priors for the Bayesian phylodynamic analyses

| Model | Parameter | Units | Prior |
|---|---|---|---|
| BDSKY | $t_{or}$ | Ma | Unif(66,266) |
| | $\lambda_i$ | Ma$^{-1}$ | Exp(1.0) |
| | $\mu_i$ | Ma$^{-1}$ | Exp(1.0) |
| | $\psi_i$ | Ma$^{-1}$ | Exp(0.2) |
| Piecewise coalescent | $t_{root}$ | Ma | Unif(66,266) |
| | $r_i$ | Ma$^{-1}$ | Norm(0.0,0.5) |
| | $N_f$ | Ma | LogNorm(1.0,1.25) |

when analysing extant phylogenies, allowing branches to reach the boundary without becoming extinct. Modelling sampling and extinction processes separately makes our approach robust to issues such as the Signor-Lipps effect (Signor and Lipps, 1982).

Both of the models described were implemented using the phylogenetic inference software BEAST 2 (Barido-Sottani et al., 2018; Bouckaert et al., 2019), using its MCMC algorithm to sample the above posterior probability distributions conditional on each of the trees. Each BDSKY and coalescent chain was run until the effective sample size for each model parameter was greater than 200, and therefore considered to have converged. The first 10% of each chain was discarded as burn-in prior to further analysis. Subsequent data processing and figure plotting was carried out in R (R Core Team, 2022). All relevant BEAST 2 input files and R scripts are available in the electronic supplement.

## Results

The results of the coalescent analyses conducted using the 16 phylogenies are summarised in Figure 1, with the corresponding numerical estimates shown in the Supplementary Tables. There is clear variation in the median diversification estimates obtained, and

the width of the error bars, between the analyses based on different clades, phylogenies, and branch lengths. However, most of the exponential coalescent models indicate a small but negative diversification rate in dinosaurs in the Coniacian–Maastrichtian (69.8% of posterior negative for Lloyd1, 96.5% for Benson1, 96.8% for Benson2, 94.1% for Lloyd2), and for some, this is also true of the preceding Aptian–Turonian (99.2% of posterior negative for Lloyd1, 68.2% for Benson1, 68.9% for Benson2, 100.0% for Lloyd2). Positive diversification rates are generally favoured in all other time bins, with the exception of the Early Jurassic (94.4% of posterior negative for Lloyd1, 98.9% for Benson1, 89.7% for Benson2, 99.7% for Lloyd2). In the full phylogenies and all three subclades, diversification rate uncertainty is highest in the Early–Mid Triassic and tends to decrease over the Mesozoic. There is most disagreement between the phylogenies for the sauropods, with the smallest (Lloyd1) phylogeny showing opposite diversification trends to the other three. However, for all of the phylogenies, most of the posterior probability lies on a strong Coniacian–Maastrichtian decline for the clade (100.0% of posterior negative for Lloyd1, 90.2% for Benson1, 90.2% for Benson2, 100.0% for Lloyd2). For theropods, a small but negative diversification rate is inferred immediately before the K-Pg boundary (99.4% of posterior negative for Lloyd1, 99.9% for Benson1, 99.9% for Benson2, 97.0% for

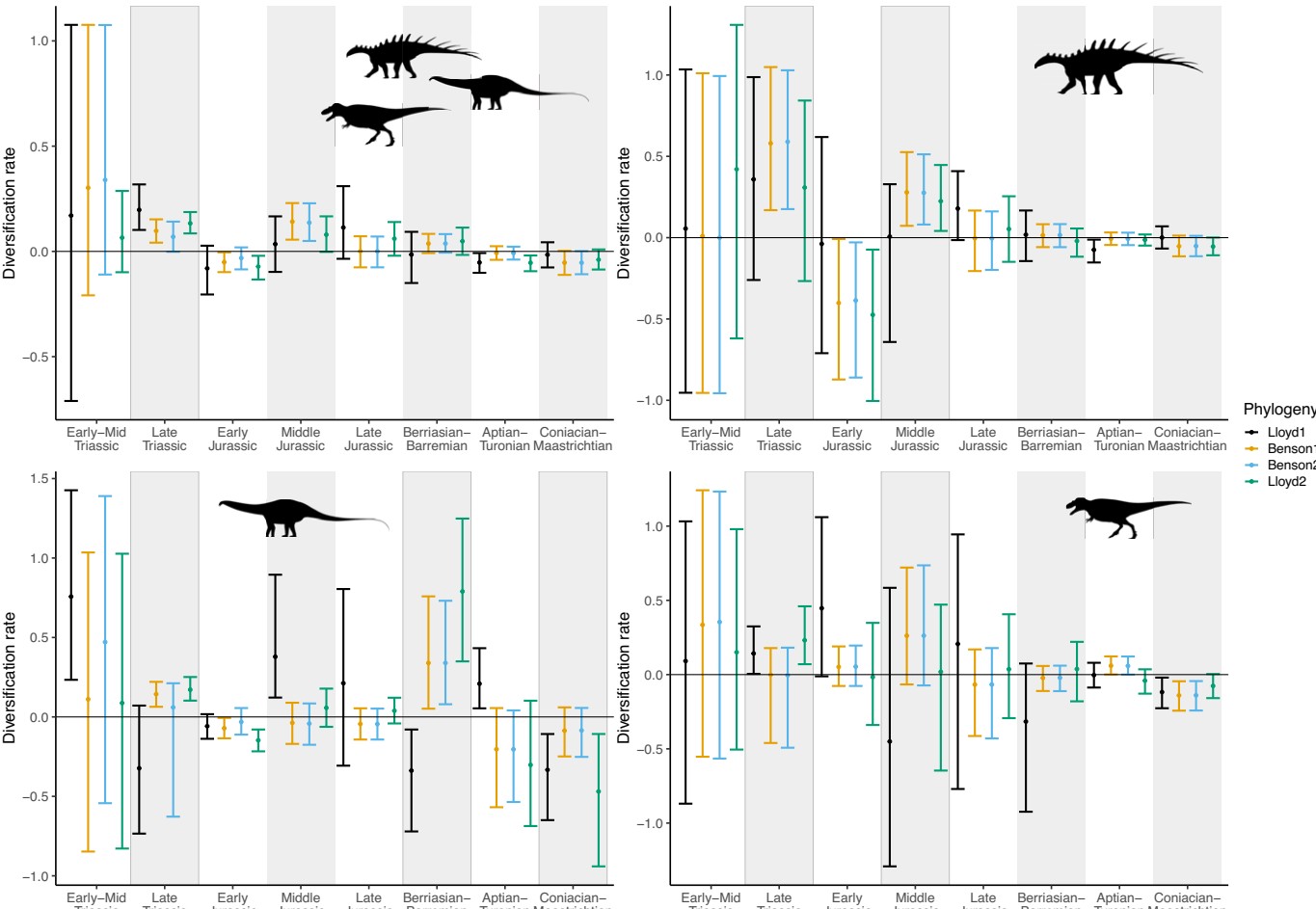

**Figure 1.** Diversification rates estimated using the piecewise-exponential coalescent model. Time moves forwards from left to right along the *x*-axis, with the K-Pg boundary at the end of the Coniacian–Maastrichtian bin. Estimates are shown for each of four phylogenies, ordered from smallest to largest. Points show the median values, and error bars indicate 95% highest posterior density. Dinosaur silhouettes for Ornithischia (top right), Sauropodomorpha (bottom left) and Theropoda (bottom right) are from Phylopic.

Lloyd2). For the ornithischians, the two Benson and larger Lloyd trees all indicate a latest Cretaceous decline (94.5% of posterior negative for Benson1, 94.6% for Benson2, 97.3% for Lloyd2), whereas the smaller Lloyd tree suggests no substantial change in diversity during this interval (49.2% of posterior negative).

The results of the birth-death analyses are summarised in Figures 2 and 3, which show the piecewise-constant estimates of diversification and sampling rates, respectively, from each of the phylogenies, and Supplementary Tables, which provide the estimated parameter values. There is less variation in the BDSKY results between the different subclades, and based on the different tree topologies, than in the coalescent results.

The most apparent pattern is that all of the models have much greater uncertainty on diversification rates in the final time bin, the Coniacian–Maastrichtian (Figure 2). This is coupled with an increase in the inferred sampling rates during this interval (Figure 3). In the full phylogenies and all three subclades, the scale of this effect decreases with increasing phylogeny size.

Despite this, in the BDSKY analyses, all four phylogenies place most posterior probability on a positive diversification rate for dinosaurs in the latest Cretaceous (90.0% of posterior positive for Lloyd1, 98.0% for Benson1, 98.1% for Benson2, 99.9% for Lloyd2). In all three subclades, it is more unclear as to whether

diversification was positive or negative, or simply constant, prior to the K-Pg boundary. All of the models appear to favour positive diversification in the Late Jurassic (99.9% of posterior positive for Lloyd1, 100.0% for Benson1, 100.0% for Benson2, 100.0% for Lloyd2), and also in the Aptian–Turonian (98.4% of posterior positive for Lloyd1, 98.4% for Benson1, 97.4% for Benson2, 99.2% for Lloyd2).

## Discussion

In this study, we characterise dinosaur diversification using two different phylodynamic models: the birth-death-sampling (BDSKY) and coalescent skyline models. The coalescent model recovered a downturn in diversity during the latest Cretaceous with a posterior probability of 97% using the Benson phylogenies, and a posterior probability of 94% using the larger Lloyd phylogeny (Figure 1). The BDSKY model instead inferred an increase in dinosaur diversity in the latest Cretaceous with a posterior probability of more than 98% based on these three largest phylogenies (Figure 2). Our results therefore span the range of diversification estimates obtained using other methods in previous literature. The difference in results we obtained using the two phylodynamic models can be linked directly to the different assumptions they

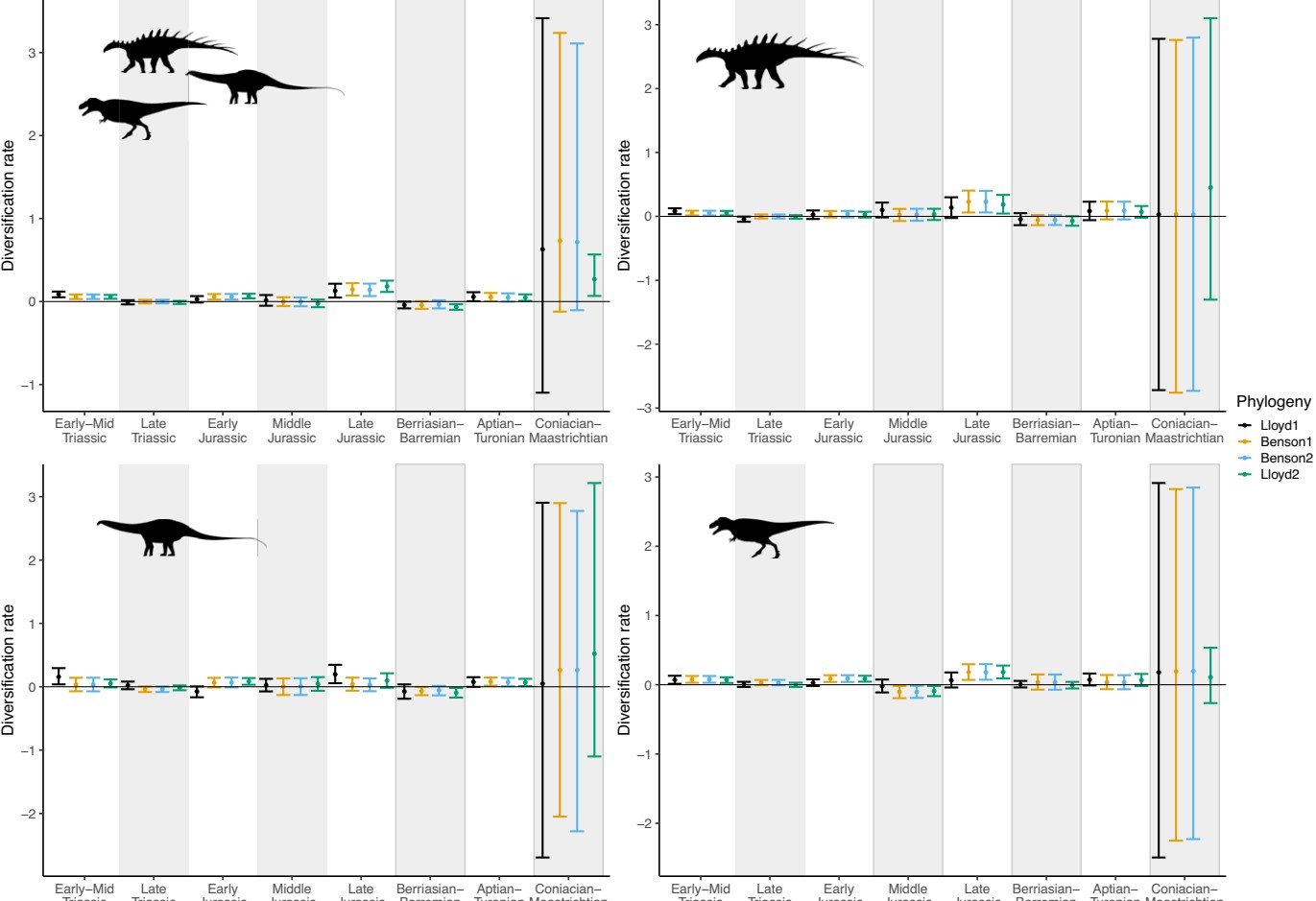

**Figure 2.** Diversification rates estimated using the piecewise-constant fossilised birth-death skyline model. Time moves forwards from left to right along the *x*-axis, with the K-Pg boundary at the end of the Coniacian–Maastrichtian bin. Estimates are shown for each of four phylogenies, ordered from smallest to largest. Points show the median values, and error bars indicate 95% highest posterior density. Dinosaur silhouettes for Ornithischia (top right), Sauropodomorpha (bottom left) and Theropoda (bottom right) are from Phylopic.

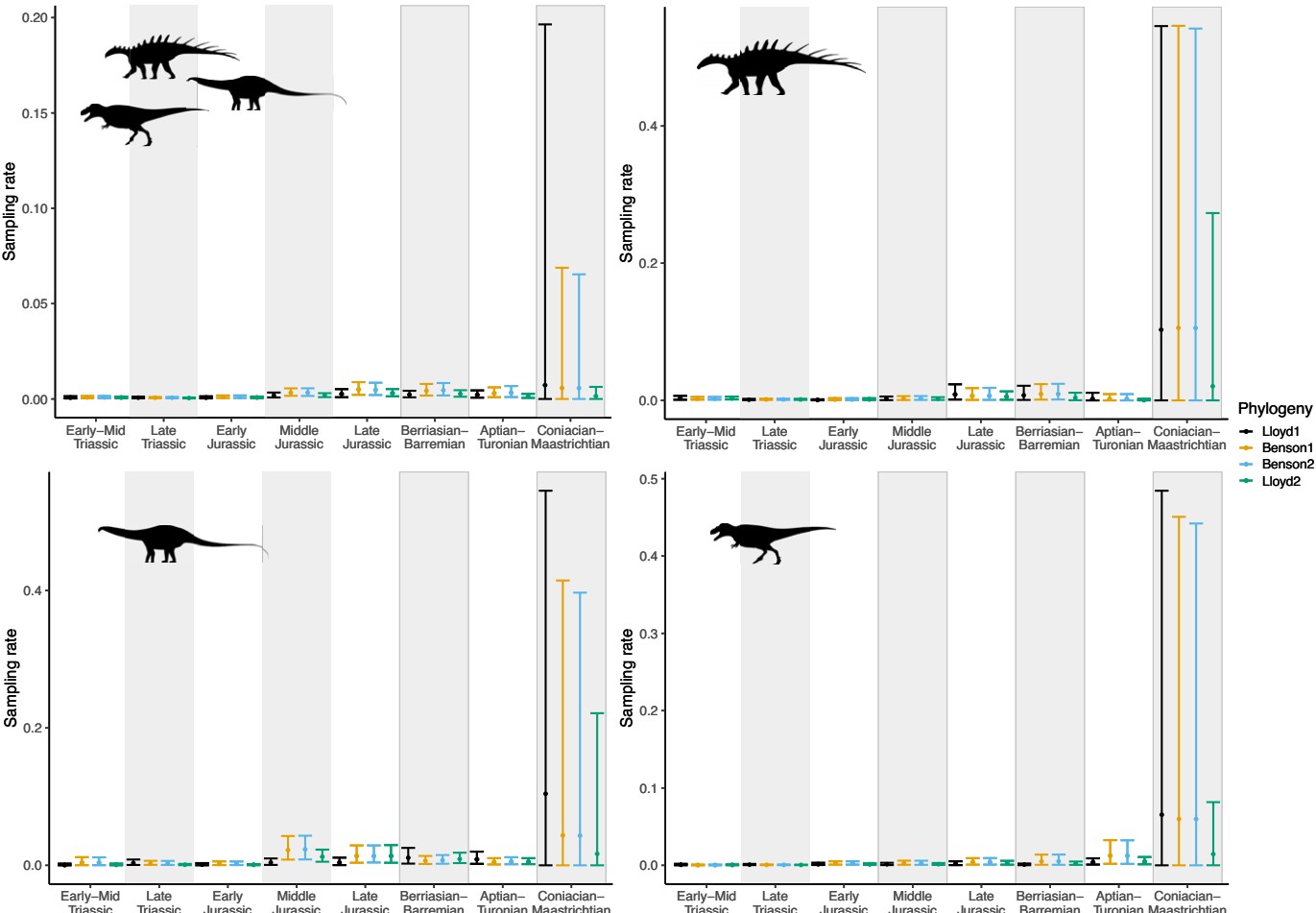

**Figure 3.** Sampling rates estimated using the piecewise-constant fossilised birth-death model. Time moves forwards from left to right along the *x*-axis, with the K-Pg boundary at the end of the Coniacian–Maastrichtian bin. Estimates are shown for each of four phylogenies, ordered from smallest to largest. Points show the median values, and error bars indicate 95% highest posterior density. Dinosaur silhouettes for Ornithischia (top right), Sauropodomorpha (bottom left) and Theropoda (bottom right) are from Phylopic.

make, highlighting that modelling decisions, whether conscious or unconscious, can qualitatively impact estimated diversification trajectories.

First, although the number of species through time changes stochastically in the birth-death model, this change is a deterministic (exponential) function of the parameters in the coalescent model. This contrast should have the largest impact when the number of species is very low, meaning both early in the history of the clade, and just prior to total extinction. As a result, we might expect to see the greatest difference between the model results in the first and last time bins (so this could be considered an "edge effect"). This effect may be contributing to the stark difference in our diversification estimates for the Coniacian–Maastrichtian time bin between the two models.

Second, the coalescent model assumes no relationship between species richness dynamics and the number and times of the samples: practically, each fossilisation event is treated as an independent phenomenon. The birth-death model instead treats sampling as a process, parameterised in the model as a rate (which is constant within each time bin). This rate is dependent upon the number of lineages, and therefore species, present at that time. The Coniacian–Maastrichtian is the most heavily sampled interval in our dataset (Supplementary Figure 1; Close et al., 2017; 2019), and for the coalescent analyses, this results in relatively narrow HPD intervals

on these diversification estimates in comparison with the other time bins (Figure 1). In contrast, in all of the birth-death-sampling analyses, we see drastically elevated uncertainty in estimated diversification rates for the Coniacian–Maastrichtian (Figure 2), with corresponding high uncertainty in the sampling rate (Figure 3). The birth-death-sampling model cannot discern whether this increased density of fossil sampling is due to a higher sampling or diversification rate, as reflected in the posterior distributions. However, we also see that the width of the HPD intervals for estimated diversification and sampling rates in the Coniacian–Maastrichtian decrease with increasing phylogeny size (Figures 2 and 3). Providing the birth-death-sampling model with more data therefore seems to reduce the uncertainty in our parameter estimates; increasing the size of the phylogenies used to conduct our skyline analyses may therefore allow us to infer more accurate diversification estimates in future.

The birth and death rates estimated in our birth-death-sampling models are, in some cases, fairly high in relation to previous estimates: median values for some phylogenies reach more than three events per lineage per million years (Supplementary Table 1), whereas Henao Diaz et al. (2019) estimated 0.02 to 1.54 events across a variety of clades, and Lloyd et al. (2017) estimated 0.94 events for dinosaurs. However, these models generally estimated relatively low diversification rates (Figure 2), with birth and death

rates closely coupled in all time intervals, except the Coniacian-Maastrichtian, for all phylogenies (Supplementary Table 1). This coupling has previously been observed in other analyses of diversification in the fossil record (Alroy, 2008; Henao Diaz et al., 2019; Černý et al., 2022), and suggests that while diversification can probably be estimated fairly reliably, disentangling speciation and extinction rates is more difficult.

Although the birth-death-sampling results suggest that all three dinosaur clades maintained their diversity or underwent slightly positive diversification throughout the Cretaceous, the coalescent results suggest that dinosaur diversity may have been in decline from the Aptian to Maastrichtian (Figures 1 and 2). The coalescent models suggest that although ornithischians and theropods may only have experienced a Coniacian-Maastrichtian decline, sauropodomorphs may have had negative diversification rates between the Aptian and Maastrichtian. This is consistent with other previous studies which found that ornithischians may have had higher diversification rates in the Cretaceous, particularly hadrosaurs and ceratopsids (Lloyd et al., 2008; Barrett et al., 2009; Sakamoto et al., 2016), alongside a previously reported reduction in the number of sauropodomorph fossils into the Late Cretaceous (Barrett and Upchurch, 2005; Mannion et al., 2011; Starrfelt and Liow, 2016). Positive Late Jurassic diversification rates in all clades suggested by the birth-death models correspond to an observed peak in local dinosaur richness (Close et al., 2019).

Previous attempts to include sampling bias in methods that estimate diversification have used proxy data, measurable variables thought to correlate with less tangible factors affecting diversity in the fossil record. The number of dinosaur-bearing geological formations is a commonly used example, thought to correlate with the amount of terrestrial rock outcrop area for each geological stage, which is expected to be a strong influence on the age distribution of collected fossils (Wang and Dodson, 2006; Barrett et al., 2009; Lloyd, 2011; Upchurch et al., 2011; Starrfelt and Liow, 2016). Sakamoto et al. (2016) used proxy data as a covariate in their phylogenetic GLMMs, and some modelling approaches have used various types of proxy data to try and extract "residual" patterns of dinosaur diversity (Barrett et al., 2009; Lloyd, 2011). However, simulation studies have demonstrated that residual modelling, particularly using geological proxies, may degrade the biological signal in the data rather than eliminating bias (Brocklehurst, 2015; Sakamoto et al., 2017; Dunhill et al., 2018). A proxy-based approach also fails to acknowledge the wide variety of biases that affect the fossil record (Raup, 1976), such as Lagerstätten effects (Walker et al., 2020), preservation biases based on morphology (Brusatte et al., 2015; Benson, 2018), and "dark data" in museums and private collections (Marshall et al., 2018). Aside from rock outcrop area, the geography of fossil collection is also greatly driven by political and socio-economic factors (Raja et al., 2022). For dinosaurs specifically, the known record is highly concentrated in North America (Hurlbert and Archibald, 1995; Le Loeuff, 2012; Brusatte et al., 2015; Chiarenza et al., 2019; Dean et al., 2020; Han et al., 2022), although the strength of this bias has reduced, and increasingly been accounted for, over time (e.g. Close et al., 2019). Fossil abundance metrics have also been used, but even these are an imperfect proxy for sampling bias, especially when integrating data from phylogenies and fossil databases that do not contain the same taxa.

The approaches to sampling used by our coalescent and birth-death-sampling models are also not a perfect fit for the true nature of the fossil record, and violations of both models' sampling assumptions may be biasing our results (e.g. Karcher et al., 2016).

However, methods for estimating diversity which attempt to mechanistically model sampling in a more realistic way will likely be a necessary step in unravelling how fossil record bias impacts our understanding of biodiversity in deep time (Brusatte et al., 2015; Starrfelt and Liow, 2016; Černý et al., 2022). Aside from this, there are additional ways in which the approach we used might be improved in future. Both of our models expect that sampling is randomly distributed across co-existing lineages, an assumption held by most approaches to estimating diversity in the fossil record, but which is not true (Hurlbert and Archibald, 1995; Wang and Dodson, 2006; Benson, 2018; Černý et al., 2022). Multi-type models may be used to allocate species to categories with different sampling parameters (Kühnert et al., 2016), however more thought is needed on how best to assign species to discrete categories. Piecewise-constant models, as used in this paper, may be vulnerable to inaccuracies when large fluctuations in rates are present within a single bin (similarly to TRiPS (Starrfelt and Liow, 2016)), and more understanding of how change time choice is important for achieving convergence and obtaining meaningful rate estimates is needed (e.g. Allen et al., 2024). Others have also commented on the sensitivity of models to input parameters and priors more broadly (Starrfelt and Liow, 2016; O'Reilly and Donoghue, 2020; Černý et al., 2022), and highlighted the importance of making careful, informed decisions when choosing analyses and carrying out model adequacy tests when possible. Careful prior choice is also required to avoid rate non-identifiability (Smiley, 2018; Louca and Pennell, 2020; Černý et al., 2022), although piecewise-constant methods may be more robust to this problem than those which generate continuous curves (Legried and Terhorst, 2022; Truman et al., 2024).

Previous authors have commented on the necessity of continuing to collect new fossils to improve our knowledge of dinosaur evolutionary dynamics (e.g. Benson, 2018; Bonsor et al., 2020; Černý and Simonoff, 2023), to which we would add that there are also many ways in which we could make better use of the fossils and data we already have. With further model development, full Bayesian phylodynamic inference of the tree and model parameters may become possible, allowing estimation of evolutionary rates across uncertainty in the topology and branch lengths of the phylogeny. This would address issues around the sensitivity of results to tree shape in currently available methods (shown here but also by Bonsor et al. (2020) and Sakamoto et al. (2021)). Such an approach could also allow for the inclusion of more data, such as the incorporation of more fossil age information (Stadler et al., 2018; Warnock et al., 2020), and utilising fossils both with and without character data (Andréoletti et al., 2022). Larger phylogenies may also enable such a model to infer evolutionary rates at a finer temporal resolution. Between the results presented here and the aforementioned potential for future improvement, it is clear that phylodynamic models can provide important insights into macro-evolutionary processes.

## Conclusions

The trajectory of non-avian dinosaur diversification prior to their demise at the K-Pg boundary has been fiercely debated. Here, we apply two phylodynamic models to dinosaur phylogenies, to investigate the influence of sampling assumptions on estimates of evolutionary rates. Our birth-death-sampling skyline model results do not support a Cretaceous downturn in dinosaur diversity, whereas the piecewise-exponential coalescent model results do. This

disparity in results indicates that fundamental differences in model design, especially with respect to sampling, can have a dramatic influence on estimates of diversification. It also highlights the importance of understanding model assumptions more broadly, providing context for results and facilitating comparison between models. Future work examining the fit of existing phylodynamic models to palaeontological datasets will help to illuminate whether one model should be favoured above the other and highlight areas for future model development.

**Open peer review.** To view the open peer review materials for this article, please visit http://doi.org/10.1017/ext.2024.5.

**Supplementary material.** The supplementary material for this article can be found at http://doi.org/10.1017/ext.2024.5.

**Data availability statement.** All files and code necessary to run these analyses are available in the associated Zenodo repository https://doi.org/10.5281/zenodo.10996357. A Taming the Beast tutorial (Barido-Sottani et al., 2018) explaining how to apply these methods (using these analyses as a case study) is also available at https://taming-the-beast.org/tutorials/.

**Acknowledgements.** We thank members of the Computational Evolution group, particularly Louis du Plessis, for helpful discussion, and Graeme Lloyd for assistance with using the phylogenies. We thank the anonymous reviewers for their comments, which helped to improve the manuscript. All BEAST 2 analyses were conducted using the ETH Zurich Euler cluster. We thank those who have contributed to the Paleobiology Database data used in this study. This is Paleobiology Database Official Publication No. 481. We also thank the artists who contributed the Phylopic images used in our figures: Matt Dempsey, Jaime Headden, and Iain Reid.

**Author contribution.** R.C.M.W., T.S. and T.G.V. developed the study concept. M.V.V.O. and B.J.A. wrote the code, with help from R.C.M.W. and T.G.V. B.J.A. and M.V.V.O. conducted the analyses, and B.J.A. made the figures. B.J.A., R.C.M.W. and T.G.V. wrote the manuscript. All authors edited the manuscript and approved its final version.

**Financial support.** This work was supported by the Amgen Scholars Programme (MVVO) and an ETH+ grant (BJA, "BECCY").

**Competing interest.** The authors declare no competing interest exist.

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
