## [Editor Report]

EXT-22-0017 „Mechanistic phylodynamic models do not provide conclusive evidence that dinosaurs were in decline before their final extinction”

While reviewer #1 was happy about your submission and suggested only minor revisions (mostly regarding colloquial language), reviewer #2 was more critical along diverse routes. Below, I provide my take on reviewer’s #2 critique:

Your phylogenetic dataset may not be up to date (a view shared with reviewer #1) and simple tree-averaging might not be justified. I understand that it is beyond the scope of this paper to produce a new supertree of dinosaurs. However, it would indeed be good to use the state-of-the-art tree and I know too little of dinosaurs to judge whether a more recent tree is available. Even if it is not, it may indeed be better to give more weight to a more recent tree in case of conflicting results. The suggestion to see how results depend on phylogenies vs. phylodynamic models is a good one and should be pursued by the authors. 

Regarding novelty, I share the reviewers concern that the basic conclusion have been reached many time before. Quantifying the decline or absence thereof might be a worthwhile endeavor.

---

## [Editor Report]

This is a well-written and thought-provoking manuscript arguing that the available data (and sampling models) are simply not sufficient to judge if non-avian dinosaurs were in decline prior to their extinction at the end of the Cretaceous period.

The reviews suggest (very) minor revisions and I second their evaluation. Actually, while a short comment on Louca & Pennell (2020, Nature 580:502-505) could be useful, I ask the authors to refrain from a lengthy discussion about the use of timetrees in diversification dynamics (suggested by reviewer #1). The Signor-Lipps effect on the branch ends, however, is a valid point that should be tackled. Perhaps this is already considered in the model.

The only major comment of reviewer #2 regards the temporal binning of the model (change times). I second that this appears a bit ad hoc and requires justification. 

l. 91-92: “infer what we do not know from what we do.” I don’t understand the “from what we do” subphrase

Figs. 1-3: The dark grey vertical stripes should be lighter. The black (Lloyd1) bars are almost not visible. Yes transforming the y axis in Fig. 3 might be useful, although a square-root transformation might work better than a log transformation as you seem to have several 0 values.

---

## [Editor Report]

The authors have successfully dealt with all pending issues and the manuscript is now in perfect shape to be accepted for our journal. Thank you for this important study.